# A silicon photoanode protected with TiO₂/stainless steel bilayer stack for solar seawater splitting

Shixuan Zhao[1,2], Bin Liu[1,2], Kailang Li[1,2], Shujie Wang[1,2,3], Gong Zhang [1,2], Zhi-Jian Zhao [1,2,4,5], Tuo Wang [1,2,4,5] ✉ & Jinlong Gong [1,2,3,4,5] ✉

Photoelectrochemical seawater splitting is a promising route for direct utilization of solar energy and abundant seawater resources for H₂ production. However, the complex salinity composition in seawater results in intractable challenges for photoelectrodes. This paper describes the fabrication of a bilayer stack consisting of stainless steel and TiO₂ as a cocatalyst and protective layer for Si photoanode. The chromium-incorporated NiFe (oxy)hydroxide converted from stainless steel film serves as a protective cocatalyst for efficient oxygen evolution and retarding the adsorption of corrosive ions from seawater, while the TiO₂ is capable of avoiding the plasma damage of the surface layer of Si photoanode during the sputtering of stainless steel catalysts. By implementing this approach, the TiO₂ layer effectively shields the vulnerable semiconductor photoelectrode from the harsh plasma sputtering conditions in stainless steel coating, preventing surface damages. Finally, the Si photoanode with the bilayer stack inhibits the adsorption of chloride and realizes 167 h stability in chloride-containing alkaline electrolytes. Furthermore, this photoanode also demonstrates stable performance under alkaline natural seawater for over 50 h with an applied bias photon-to-current efficiency of 2.62%.

Photoelectrochemical (PEC) water splitting utilizes photoelectrical components to convert sustainable solar energy into hydrogen, which alleviates the reliance on non-renewable fossil fuels[1]. Purified water is usually used to prepare electrolytes for PEC water splitting in the laboratory and commercial water electrolyzers, whereas a large proportion of the global population is still in shortage of drinking water. In contrast, seawater, which occupies around 96.5% of earth's water, is a desirable and widespread natural resource for water electrolysis. Currently, the main routes to produce hydrogen from seawater include indirect and direct seawater splitting. Although indirect seawater splitting, such as traditional water electrolysis

coupled with seawater reverse osmosis (SWRO), integrates several mature technology, it also increases the expenditure and the complexity of the system. While the latter approach is economically more feasible for current technology[2], from a long term perspective of view, direct seawater splitting coupled with solar or wind energy may avoid the cost and complexity of seawater purification process, which is suitable for application scenarios where space is limited such as offshore platforms or ships[3]. Compared with direct seawater electrolysis conducted through electrolyzers powered by photovoltaic systems, PEC seawater splitting could utilize solar energy directly without additional micro-grids or AC-DC devices, which may

¹School of Chemical Engineering and Technology; Key Laboratory for Green Chemical Technology of Ministry of Education, Tianjin University, Tianjin 300072, China. ²Collaborative Innovation Center of Chemical Science and Engineering (Tianjin), Tianjin 300072, China. ³Joint School of National University of Singapore and Tianjin University, International Campus of Tianjin University, Binhai New City, Fuzhou 350207, China. ⁴Haihe Laboratory of Sustainable Chemical Transformations, Tianjin 300192, China. ⁵National Industry-Education Platform of Energy Storage, Tianjin 300350, China. ✉e-mail: wangtuo@tju.edu.cn; jlgong@tju.edu.cn

further simplify the process of $H_2$ production from seawater. In terms of photocatalysis seawater splitting, although some researchers recently investigated that indium gallium nitride photocatalyst achieved solar to hydrogen efficiency of 6.2% in a large-scale photocatalytic water-splitting system, this technology also faces several challenges, such as gas separation of $H_2$ and $O_2$ and precious metal photocatalyst."

However, the complicated salinity in seawater presents a severe obstacle to the stable operation of photoelectrodes as well as entire photoelectrolytic cells[2]. PEC seawater splitting is confronted with complex challenges, such as chloride oxidations, corrosion, precipitation, membrane fouling, etc[4–6]. Compared with photocathodes, photoanodes face more intractable problems derived from corrosion and the competition between chloride oxidation and oxygen evolution reaction (OER)[7]. Thus, cocatalysts or protective layers are needed between the photoanode and electrolyte for anticorrosion purposes to enable stable PEC seawater splitting operations[8,9]. For instance, the $MoO_3$ layer, constructed with surface doping engineering, served as the barrier layer for corrosive ions, while dual-doping of B and Mo on $BiVO_4$ enhanced the utilization efficiency of photogenerated holes and achieved the highest HClO production in seawater splitting[10]. However, chlorine is corrosive and difficult to store and transport, which makes it a less desirable product on the anode. To suppress chloride oxidation, the alkaline condition is preferable because the cocatalysts can reveal the 480 mV overpotential superiority, inhibiting chlorine chemistry[11]. Nevertheless, the present design of cocatalysts or protective layers on photoanodes are still under intensive research for higher stability and overall hydrogen production rate.

Significant progress in saline water electrolysis has been reported, with several novel anode structures for anticorrosion, including the $Cl^-$ blocking layer[12] as well as the polyatomic anion-rich catalysts[13]. These protective configurations have also been proven to be effective on photoanodes for the PEC system. For instance, efficient catalysts in seawater electrolysis, including $MnO_x$[14], cobalt phosphate[15], $NiFeO_x$[16], and $RhO_2$[17], have been investigated on PEC seawater splitting and acquired remarkable performances. Recently, stainless steels have been studied as non-noble electrocatalysts for water splitting on account of high-activity transition metals and remarkable conductivity[18–20]. Although stainless steel has presented its potential performance towards water electrolysis and anticorrosion, its application as the protective cocatalyst on photoelectrodes in PEC systems is still restricted by the incompatible preparation methods with semiconductor based photoelectrodes, as well as the resultant interfacial defects, and its parasitic light absorption.

The loading of cocatalysts on photoelectrodes in PEC systems is more challenging compared with electrolysis systems because the semiconductor substrates are more vulnerable to the surface and interfacial damages during the introduction of cocatalysts, which may generate interfacial defects that result in the recombination of photogenerated minority carriers[21,22]. Thin film deposition techniques, such as physical vapor deposition, are effective and convenient methods for preparing cocatalysts for photoelectrodes. In comparison with bulk alloy, stainless steel thin films produced by sputtering are prone to form finer grain size and amorphous structure, which results in intensive pitting resistance[23]. However, the plasma damage derived from sputtering cannot be ignored for semiconductor substrates. The ion flux in the plasma exerts a negative effect on effective minority carrier lifetime for silicon heterojunction solar cells[24]. In addition, the compositions of the films may vary from bulk stainless steel targets, owning to preferential sputtering of elements[25]. Furthermore, another intractable drawback associated with stainless steel films is the dilemma between the suitable thickness for chloride resistance and optical loss[26]. Therefore, great efforts are needed to overcome the plasma damage on the interface due to sputtering and optical loss

because of opaque stainless steel films to achieve remarkable PEC performance in seawater splitting.

This paper describes the design and fabrication of a bilayer stack composed of stainless steel and titanium dioxide ($TiO_2$), serving as the catalytic and protective layer for Si photoanode, which achieves PEC seawater splitting performance. The compact stainless steel top layer presents effective anticorrosion in seawater and efficient OER catalytic ability, while the $TiO_2$ bottom layer effectively prevents the defect states from plasma damage during sputtering of stainless steel. In consequence, this back-illuminated Si photoanode achieves a 3.65% applied bias photon-to-current efficiency (ABPE) and a long-term stability of about 167 h under chloride-containing alkaline electrolytes. The measured $H_2$ production rate of cathode in this PEC system reaches up to 600 $\mu$mol $h^{-1}$ $cm^{-2}$ in alkaline natural seawater, which stands out among previous photoanodes operating in seawater splitting.

## Results

### Construction of silicon photoanode with stainless steel and $TiO_2$ stack as a protective cocatalyst layer

The Si photoanode (Supplementary Fig. 1) was fabricated by depositing $TiO_2$ and stainless steel films on a heterojunction Si substrate (ITO/$n^+$-a-Si/a-Si/c-n-Si/a-Si/$p^+$-a-Si/ITO, denoted as HJ-Si). The $TiO_2$ layer was deposited by atomic layer deposition (ALD) with an optimized thickness of 10 nm. Stainless steel films (denoted as SS) were fabricated by direct current (DC) magnetron sputtering with an AISI 316 L target (Figs. 1a and 1b). In comparison with HJ-Si/SS, $TiO_2$ passivated photoanode (HJ-Si/$TiO_2$/SS) presents a higher photocurrent density with a much steeper increase (Supplementary Fig. 2), which renders an ABPE of 3.39% for HJ-Si/$TiO_2$/SS, 2.7 times that of HJ-Si/SS (ABPE 1.25%, Fig. 1c). The improved PEC performance could be attributed to the fact that $TiO_2$ effectively passivates the defects on the ITO layer of HJ-Si induced from the plasma damage as well as the Fermi level pinning at the metal/semiconductor interface[27,28]. The thickness of $TiO_2$ presents no significant effect on the stability of Si photoanode when its thickness exceeds 8 nm, while thicker $TiO_2$ might pose a threat to the conductivity of photoelectrode. Thus, considering both the conductivity and protection, the $TiO_2$ layer with 10 nm is selected. Owing to the protection from $TiO_2$, relatively thick catalytic layers could be sputtered from AISI 316 L and Ni, respectively, with the same thicknesses (20 nm, denoted as SS-20 and Ni-20) to compare the OER performance (Supplementary Table 1). Meanwhile, thicker SS films (83 and 123 nm) were prepared to further optimize the seawater splitting performance (denoted as SS-83 and SS-123). Upon electrochemical reconstruction in seawater, the compact SS film transformed into Cr-NiFeOOH as OER cocatalysts and restrained the substrate from electrolytes oxidation and $Cl^-$ etching (Fig. 1d). However, according to UV-vis transmission spectra (Supplementary Fig. 3), the light transmittance decreases dramatically with a thicker SS layer, which restricts the photocurrents and ABPEs of photoanodes. To alleviate this problem, the back-illuminated configuration is adopted to relieve the dilemma between light illumination and cocatalyst loading[20,27], inspired by the profound discussion[25] about the relationship between carrier diffusion length, Si thickness ratio, and photocatalytic performance. The back-illuminated photoanode was constructed from 155 $\mu$m thick silicon substrate with an ultra-long carrier diffusion length of 1940 $\mu$m[22], which ensures abundant light absorption without sacrificing PEC photocurrents.

### Electrochemical reconstruction of stainless steel during activation

The textured photoanode indicates the compact bilayer stack of $TiO_2$/SS coated on multilayered HJ-Si (Figs. 2a and 2b), as evidenced by scanning electron microscope (SEM) and cross-sectional transmission

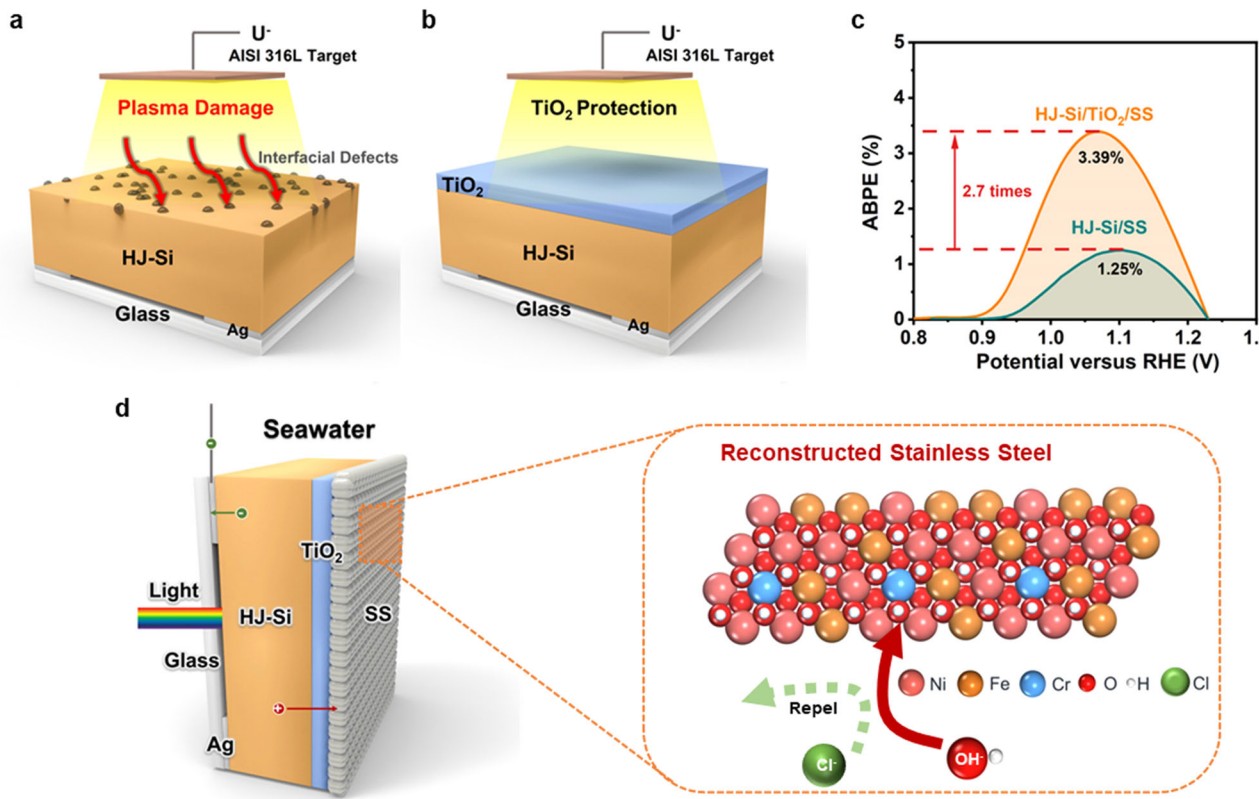

**Fig. 1 | The configuration of silicon photoanode.** A schematic illustration of synthesis process for (a) HJ-Si/SS and (b) HJ-Si/TiO₂/SS. (c) ABPE curves of HJ-Si/TiO₂/SS and HJ-Si/SS in chloride-containing alkaline electrolytes under simulated AM 1.5 G illumination. (d) Model of HJ-Si/TiO₂/SS applied in seawater splitting.

electron microscopy (TEM). The property of pristine SS film before the reaction is governed by the sputtering process. The first concern for preparation is the preferential sputtering of alloy owing to different sputtering yields of different elements, which is determined by the difference of relative atomic mass, atom surface binding energy distinction, and bombardment-induced Gibbsian segregation of targets[23]. The SS thin films were dissolved by nitrohydrochloric acid and then analyzed by inductively coupled plasma-mass spectrometry (ICP-MS), indicating that the main element proportions (Ni, Fe, Cr) in as-prepared films are similar to AISI 316 L target (Supplementary Table 2). Viewed from the top and cross-sectional SEM, SS thin films are constituted by abundant nanoscale atomic clusters, which is consistent with the Stranski–Krastanov growth mechanism (Supplementary Fig. 4). SEM element mappings reveal uniform signals, illustrating the homogeneous nature of the film (Supplementary Fig. 4). In addition, the cross-sectional TEM of HJ-Si/TiO₂/SS indicates the uniformity of TiO₂ and stainless steel layers coated on the HJ-Si substrate (Supplementary Fig. 5). However, the Bragg diffraction peak intensity of the sputtered thin films in grazing incidence X-ray diffraction decreases (Supplementary Fig. 6), which is significantly different from AISI 316 L target with a representative austenitic structure. As evidenced by the high resolution cross-sectional TEM, the pristine SS film illustrates nanocrystalline structure with discrete amorphous domains, consistent with the slight Bragg diffraction peak intensity in GIXRD (Figs. 2c and 2d). The decrease in the crystallinity of sputtered films, even forming amorphous alloy films, can be attributed to the cascade collision mechanism, breaking the pristine crystalline structure of the target[29]. As a result, the nanocrystalline or amorphous structure prepared by sputtering has been reported to present intensive pitting resistance, with fewer dislocations or grain boundaries that preferentially adsorb Cl⁻ to induce the onset of corrosion[30].

However, the electrochemical reconstruction on the surface of sputtered stainless steel films occurred during activation, which is an intrinsic and common phenomenon of various OER catalysts based on transition metals[18,31]. After activation, the near-surface compositions exhibit a prominent change, with more nickel species migrating to the surface region, as evidenced by X-ray photoelectron spectroscopy (XPS). In addition, calculated from high-resolution XPS, the sum of Cr and Fe weight proportions on the surface of pristine SS is up to over 90% (Supplementary Table 3), indicating the enrichment of Fe and Cr oxides on the surface (Figs. 2e and 2g), whereas the Ni species with a low percentage remain at metallic state (Fig. 2f). In contrast, more nickel species migrate to the surface in the reconstructed SS and the peaks located at 856.0 and 873.6 eV are ascribed to the presence of Ni²⁺ oxidation state, revealing the complete oxidation of Ni (Fig. 2f)[32]. With regard to the O 1s spectrum, two deconvoluted peaks, sited at 531.7 and 529.9 eV, are assigned to O-H and metal-O respectively (Supplementary Fig. 7)[33]. In addition, SS were peeled off by ethanol solution with ultrasound-assisted extraction and analyzed by TEM. According to the HRTEM, pristine SS exposes the (111) plane of Fe₂O₃ and (110) plane of the a-Fe, whereas the activated cocatalyst is prone to expose the (002) plane of Ni(OH)₂ (revised Supplementary Fig. 8). Thus, the sputtering stainless steel film is reconstructed into the chromium-incorporated NiFe (oxy)hydroxide and the catalytic ability and stability of activated HJ-Si/TiO₂/SS are evaluated later.

Although the compositions on the surface of sputtered stainless steel films change obviously, the structure transformation between pristine and activated samples is negligible. As evidenced by cross-sectional TEM (Fig.2b), the thickness of pristine SS is near 78 nm, an approximate value to which measured by spectroscopic ellipsometer (83 nm, Supplementary Table 1), while the thickness change of SS after reconstruction is inappreciable. Before reconstruction, the heterojunction Si substrate is covered by dense TiO₂ film and the pristine SS film before activation is compact without pin-holes or defects. After reconstruction, there are no pit holes in the SS layers as well as near the

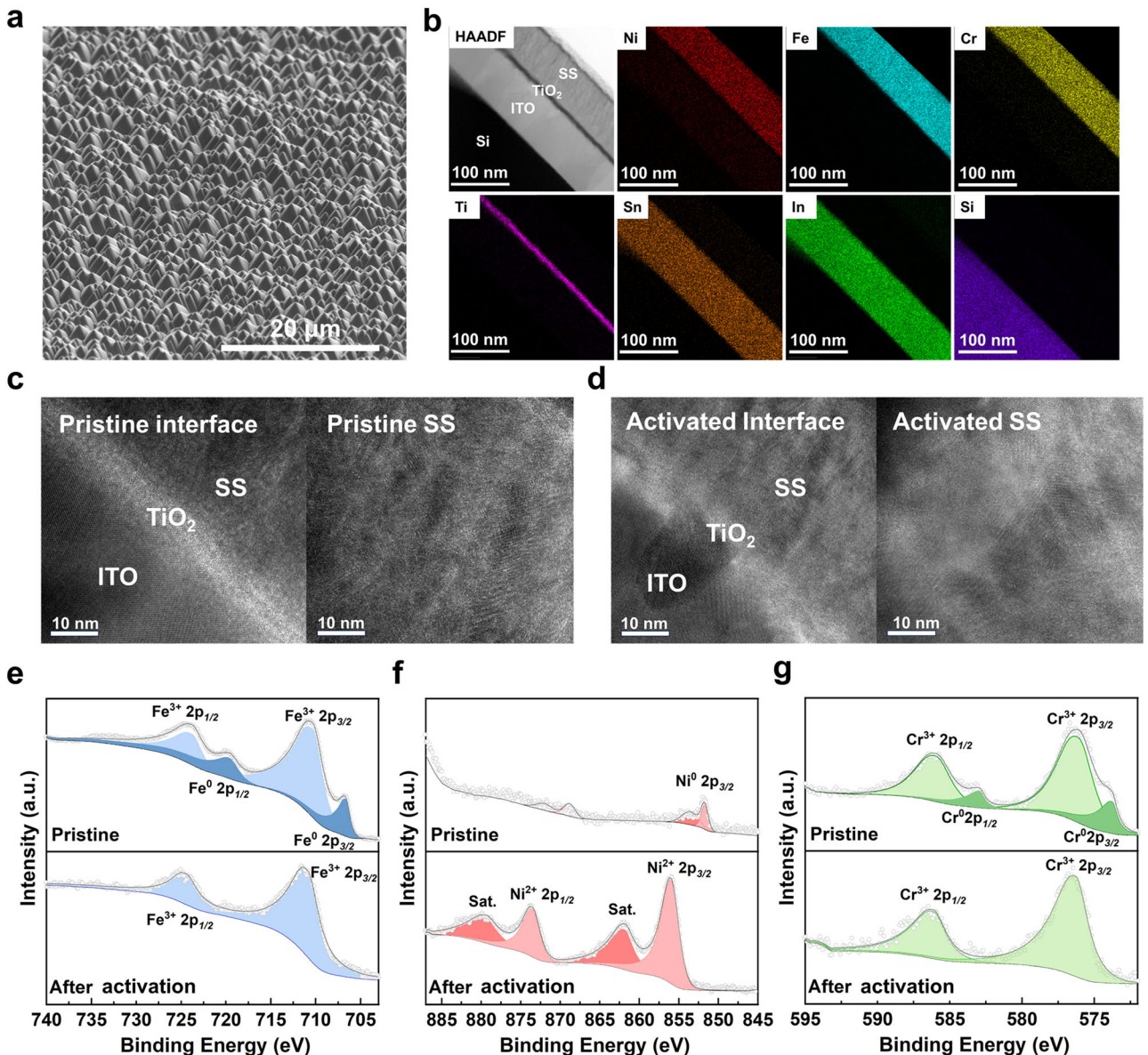

**Fig. 2 | Morphologies and chemical states of HJ-Si/TiO₂/SS.** (a) SEM images, (b) HAADF-TEM images and the corresponding elemental mapping images, high-resolution cross-sectional TEM of (c) pristine and (d) activated HJ-Si/TiO₂/SS. The high-resolution X-ray photoelectron spectroscopy of (f) Fe, (g) Ni, and (h) Cr for pristine and activated SS. The morphologies in (a–g) were characterized in HJ-Si/TiO₂/SS-83.

TiO₂/SS and Si/TiO₂ interfaces, which could be attributed to the nanocrystalline structure with discrete amorphous domains in the SS layer. By contrast, the materials with high crystallites might trigger high pit hole density between intergrain regions, which results in the contact between alkaline electrolyte and substrate[34].

## Reconstructed cocatalysts for enhanced PEC performance under chloride-containing alkaline electrolytes

The current density-potential (J-V) curves of HJ-Si/TiO₂ coated with SS-20, SS-83, SS-123, and Ni-20 were measured in 1 M KOH plus 0.5 M NaCl under simulated AM 1.5 G illumination. The drawing of quartz photoelectrochemical cells is provided (Supplementary Fig. 9). Ni is selected as the control sample due to its exceptional OER properties. When compared to samples of similar cocatalyst thicknesses (20 nm), SS-20 reveals an onset potential of 0.84 V vs. RHE at 0.1 mA cm⁻², which slightly outperforms Ni-20 (Fig. 3a). The photocurrent density of SS-20 reaches 31.65 mA cm⁻² at 1.23 V vs. RHE, approaching the saturation photocurrents. Meanwhile, the ABPE of SS-20 reaches 3.30% at 1.07 V

vs. RHE, which surpasses the one coated with Ni (Fig. 3b). The incident photon-to-current efficiency (IPCE) of our device was measured at a potential of 1.5 V vs. RHE. By integrating the measured IPCE over the standard AM 1.5 G spectrum (ASTM G173-03), a photocurrent density of 32 mA/cm² could be calculated (Supplementary Fig. 9d), which is very close to that of 34 mA/cm² at 1.5 V vs. RHE from the J-V test (Fig. 3a) under AM 1.5 G simulator. The consistency between the two sets of photocurrent density demonstrates the accuracy of our AM 1.5 G simulator in simulating sunlight, ensuring the reliability and accuracy of J-V and IPCE measurements. In addition, the electrochemical impedance spectroscopy (EIS) of different photoanodes were conducted to investigate the charge carrier transport of photoanodes, indicating that SS-20 and SS-83 efficiently reduce the carrier transfer resistances in the solid/electrolyte interface (Supplementary Fig. 10). Thus, the PEC performances of HJ-Si/TiO₂/SS are superior to HJ-Si/TiO₂/Ni owing to the electrochemical reconstruction of stainless steel thin films. In addition, different stainless steels were tested, including AISI 316 L and AISI 304, which were wide used for

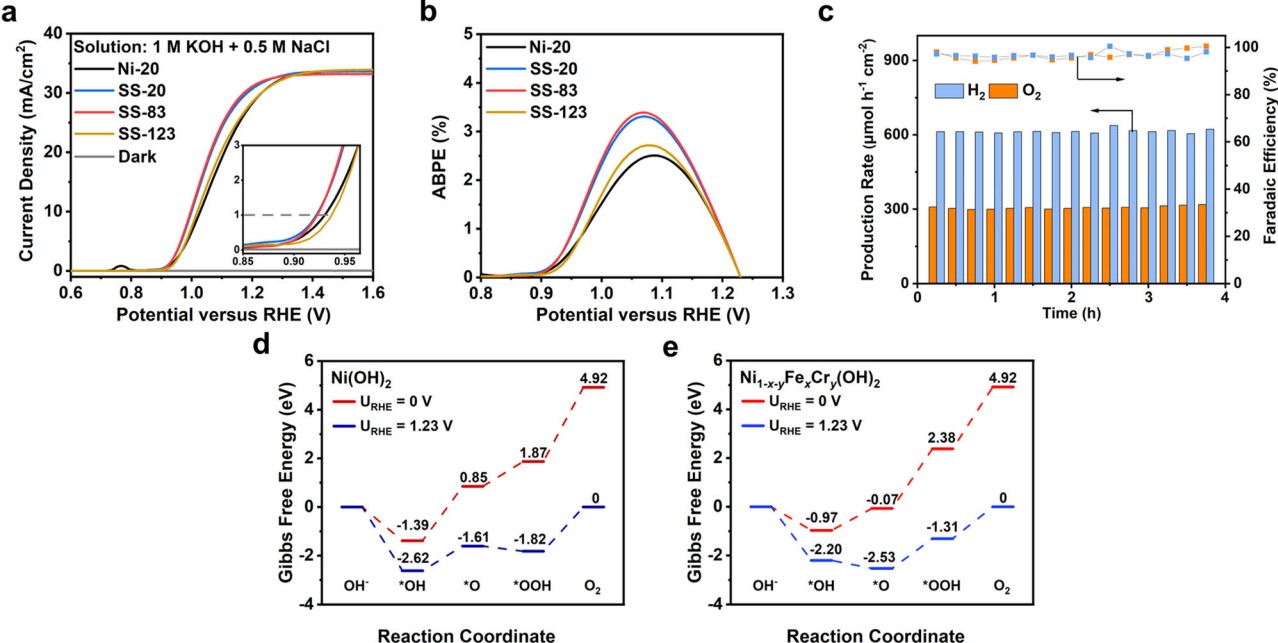

**Fig. 3 | PEC performances of fabricated silicon photoanodes under chloride-containing alkaline electrolytes without iR compensation.** (a) The current-voltage curves and (b) the corresponding ABPE curves of HJ-Si/TiO$_2$ coated with SS-20, SS-83, SS-123, and Ni-20. (c) Faradaic efficiency and production rate of H$_2$ and O$_2$ under the potential 1.5 V versus RHE of SS-83. The Gibbs free-energy diagram of (d) Ni(OH)$_2$ and (e) Ni$_{1-x-y}$Fe$_x$Cr$_y$(OH)$_2$ at 0 and. 1.23 V versus RHE. The data in (a–e) were measured in 1 M KOH + 0.5 M NaCl (pH 14) electrolyte under simulated AM 1.5 G illumination.

anticorrosion purposes (Supplementary Fig. 11). The results indicate that AISI 316 L and AISI 304 shows similar PEC performance with excellent onset potential and ABPE because these two targets possess similar compositions of Ni, Fe and Cr elements. According to high resolution XPS, the atomic ratio of Fe and Ni is around 0.4 (Supplementary Table 3), which is close to the optimal surface composition ratio in thin film Ni−Fe oxide catalysts investigated by Bell[35,36]. The decreasing proportion of Cr on the surface region is also beneficial for OER because Cr oxides are not regarded as efficient OER catalysts in comparison with Ni and Fe species.

The density functional theory (DFT) calculations were performed to explore the four-step OER mechanism on Ni$_{1-x-y}$Fe$_x$Cr$_y$(OH)$_2$ and Ni(OH)$_2$, which further facilitates the understanding of the benefits from electrochemical reconstructed stainless steel. The DFT models of Ni$_{1-x-y}$Fe$_x$Cr$_y$(OH)$_2$ and Ni(OH)$_2$ were built up and calculation details are provided in the supplement (Supplementary Fig. 12). The Gibbs free energy of OER intermediates under different potentials was calculated to investigate the rate-limiting step, which is an important parameter to evaluate the OER catalytic activity (Figs. 3d and 3e). The rate-determining step (RDS) of Ni$_{1-x-y}$Fe$_x$Cr$_y$(OH)$_2$ and Ni(OH)$_2$ is the deprotonation of the OOH*. Compared with Ni(OH)$_2$, the reaction energy of RDS over Ni$_{1-x-y}$Fe$_x$Cr$_y$(OH)$_2$ decreases from 1.82 to 1.32 eV at U$_{RHE}$ = 1.23 V (Supplementary Table 4 and Table 5), indicating the prominent promotion to OER over Ni$_{1-x-y}$Fe$_x$Cr$_y$(OH)$_2$, which is in good agreement with the excellent onset potential and ABPE of HJ-Si/TiO$_2$/SS in chloride-containing alkaline electrolytes[37].

As the SS thicknesses increase from 20 to 83 and finally 123 nm, SS-83 presents the best PEC performance under chloride-containing alkaline electrolytes with an ABPE of 3.65%, whereas there is a noticeable decay in onset potential and ABPE for SS-123 which might result from the increase in sheet resistance of SS-123 (Supplementary Table 6). The ABPE tests were repeated over four different batches of samples in order to alleviate the experiment errors (Supplementary Fig. 13). The anodic and cathodic products of the best sample, HJ-Si/TiO$_2$/SS-83, were analyzed by the gas chromatograph in a sealed H-cell with light windows (Supplementary Fig. 14) to calculate the Faradaic Efficiencies (FEs) of O$_2$ and H$_2$. The FE of O$_2$ in SS-83 remains over 96% during 10 h, which represents the apparent inhibition of ClO$^-$ production (Fig. 3c). The measured H$_2$ production rate maintains around 600 μmol h$^{-1}$ cm$^{-2}$, which is the twice production rate of O$_2$, indicating the complete utilization of carriers (Fig. 3c).

The long-term durability was explored by chronoamperometry at a bias of 1.5 V vs. RHE in 1 M KOH plus 0.5 M NaCl under simulated AM 1.5 G illumination. The current density of SS-20 remains stable for over 145 h (Fig. 4a). On the contrary, the Ni cocatalyst layer continues to flake off during a short 5 h period, presenting the corrosion vulnerability of Ni under Cl$^-$ environment. Impressively, enhanced stability was achieved from the optimized SS-83, with a photocurrent drop of only about 10% during 167 h without noticeable peeling off of the cocatalyst (Supplementary Fig. 16). This could be attributed to the fact that thicker stainless steel cocatalyst completely isolates the semiconductor substrate from the corrosive ions in seawater. By contrast, there might be voids or pit holes in the thinner SS cocatalyst (20 nm), leading to preferential adsorption of chlorides on the non-uniform regions. According to the surface morphology and J-V characteristics after the stability test, the PEC performance and structure of the photoanode show negligible changes (Fig. 4a and Supplementary Fig. 15), implying the superior anti-corrosion and catalytic ability of SS-83. After the stability test, iron or chromium partly dissolve in the electrolyte, while the contents of nickel are stable in the film (Supplementary Table 7). Upon another 168 h stability test with cyclic 8 h illumination followed by 16 h dark (open circuit) to mimic the day-night cycling operation, a stable photocurrent could be obtained by the SS-83 sample with negligible surface morphological changes (SEM images in Supplementary Fig. 17), indicating the successful isolation between electrolyte and semiconductor substrate. After the stability, iron or chromium partly dissolve in the electrolyte, while the contents of nickel are stable in the film. The polarization curves illustrate that the corrosion potentials of HJ-Si/TiO$_2$/SS are higher than HJ-Si/TiO$_2$/Ni (Supplementary Fig. 18), which indicates that sputtered SS offers

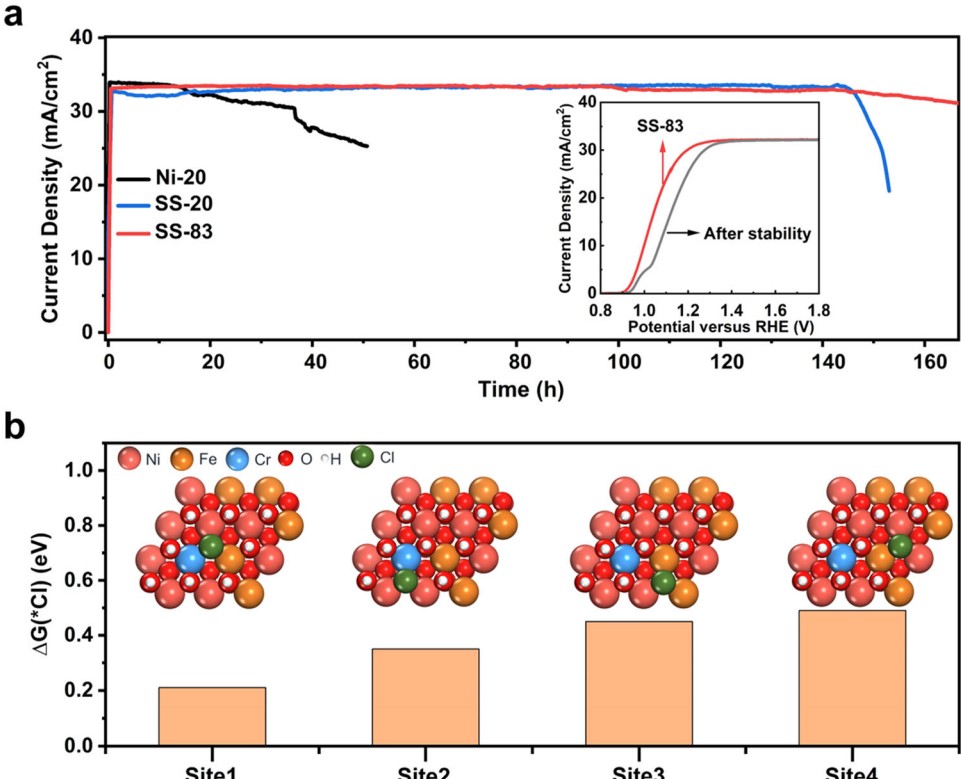

**Fig. 4 | Investigation on the effect of Cl⁻ for PEC stability without iR compensation.** (a) The stability tests and J-V curves of SS-83 before and after stability test in 1 M KOH + 0.5 M NaCl (pH 14) electrolyte under simulated AM 1.5 G illumination. (b) The adsorption geometry for four sites on $Ni_{1-x-y}Fe_xCr_y(OH)_2$ and the Cl⁻ adsorption energy differences between $Ni_{1-x-y}Fe_xCr_y(OH)_2$ and $Ni(OH)_2$. $\Delta G(*Cl) = G_{Ni1-x-yFexCry(OH)2}(*Cl) - G_{Ni(OH)2}(*Cl)$.

a thermodynamic superiority over nickel films for anticorrosion. SS-83 illustrates the highest corrosion potential and minimum corrosion current (Supplementary Table 8), which is consistent to the durability results. The corrosion rate of Ni-20 is more than 3 times higher than SS-20 and SS-83.

The DFT calculation also reveals the adsorption behaviors of chloride ions on $Ni_{1-x-y}Fe_xCr_y(OH)_2$ and $Ni(OH)_2$. According to the metal chloride-hydroxide corrosion mechanism and CER reaction pathway, Cl⁻ adsorption behaviors play an essential role in the competitive reaction on photoanode and cocatalyst stabilities. From the DFT calculation results (Fig. 4b), in comparison to $Ni(OH)_2$, Cl⁻ adsorption on each site of $Ni_{1-x-y}Fe_xCr_y(OH)_2$ is weakened, further inhibiting the first elementary reaction step of corrosion process and CER[38-40], which is a reasonable explanation for improved stability of chromium-incorporated NiFe (oxy)hydroxide on Si substrate. In addition, the chromic oxide has been utilized to adjust the local reaction environment of transition metal oxide[8], which effectively increases the local alkalinity and avoids chloride damage, achieving stable direct seawater electrolysis.

Thus, the effective OER activity and stability can be attributed to the TiO₂/SS bilayer configuration, which acts as the effective protective layer and reconstructed cocatalyst with appropriate reaction energy for OER and adsorption energy of Cl⁻.

## Photoelectrochemical performance in alkaline seawater

The photoelectrochemical performance of HJ-Si/TiO₂/SS-83 was further evaluated in seawater, including alkaline natural seawater from the BoHai Sea of China and simulated seawater with known compositions, containing Cl⁻, $SO_4^{2-}$, Br⁻, F⁻, $HCO_3^-$, $Sr^{2+}$ (Supplementary Table 9). Compared with chloride-containing alkaline electrolytes, there is an increase of overpotential at 10 mA cm⁻² for alkaline natural seawater (near 30 mV) and simulated seawater

(near 70 mV) (Fig. 5a). The ABPEs also decrease from 3.32% for 1 M KOH plus 0.5 M NaCl to 2.62% for alkaline natural seawater (Fig. 5b). However, the O₂ Faradaic efficiency still maintains over 95% during the test (Fig. 5c), indicating that the corrosion of photoanode in natural seawater is not the reason for the reduced performance. Moreover, the duration for stable seawater splitting shrinks to around 55 h (Fig. 5d), due to the severe salt precipitation on the photoanode from the natural seawater with high salinity (Supplementary Fig. 19). Although the overall performance of HJ-Si/TiO₂/SS-83 decreases, the measured H₂ production rate remains around 600 μmol h⁻¹ cm⁻² (Fig. 5c and Supplementary Fig. 20) and the calculated total H₂ production reaches 741.6 ml cm⁻² (Supplementary Table 10). The hydrogen production rate of various PEC seawater splitting systems is predicted, compared with HJ-Si/TiO₂/SS measured through GC, indicating that hydrogen production of Si photoanode with the bilayer stack is ten times more efficient than other PEC seawater splitting systems, which stands out among previously reported photoanodes on seawater splitting.

The natural seawater is multi-component with a variety of dissolved ions, resulting in competitive reactions for both photoanode and cathode. When the photoelectrodes operate in the simulated seawater without a buffer solution, the pH around the cathode will increase because of the consumption of H⁺, which results in the accumulation of insoluble hydroxides of Ca(OH)₂ and Mg(OH)₂. The addition of potassium hydroxide into the simulated seawater leads to the precipitation of Ca(OH)₂ and Mg(OH)₂ prior to the PEC test (Supplementary Fig. 21), thereby preventing the precipitation of the cathodes. In addition, $SO_4^{2-}$ has been reported to present a positive effect on stability due to the preferential adsorption of $SO_4^{2-}$, which forms the negative charge layer repulsing chloride ions[40]. However, apart from the existence of Cl⁻, other kinds of halide ions, such as Br⁻ and F⁻, also exist in the seawater, which is evidenced to accelerate catalyst

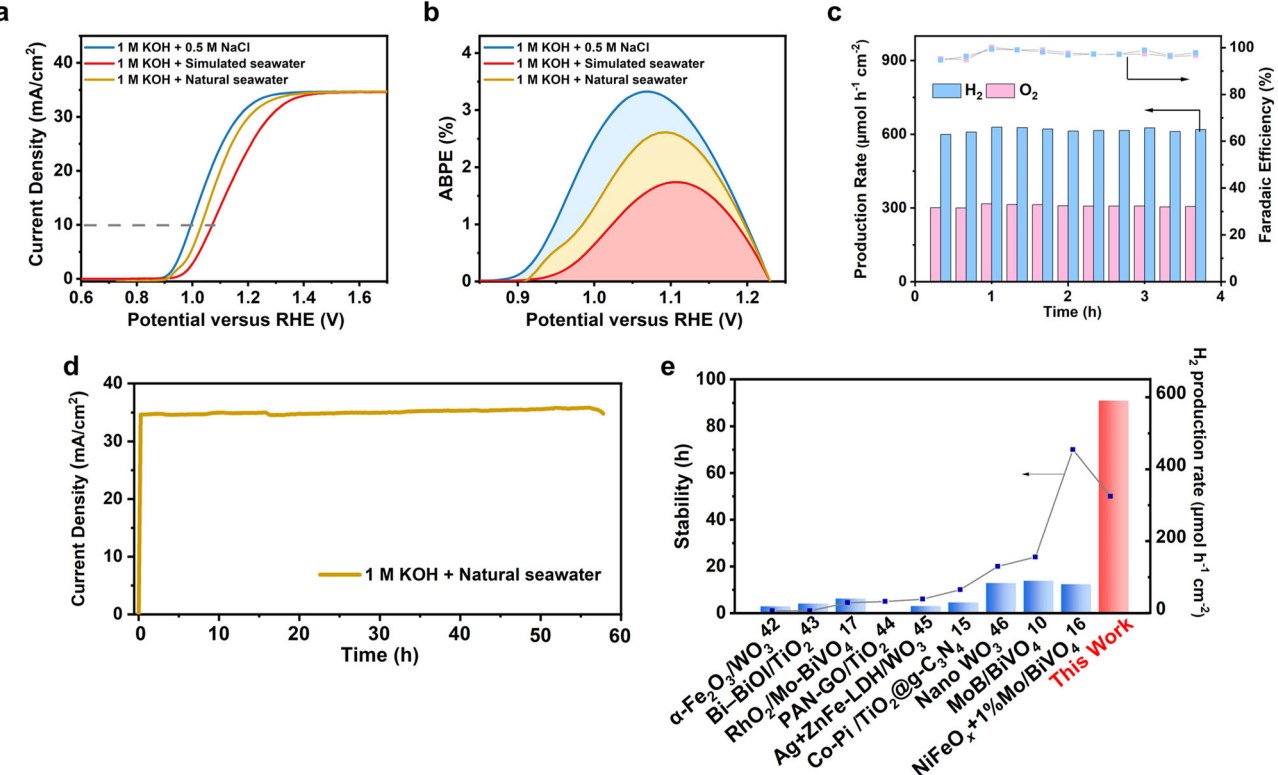

**Fig. 5 | PEC performances of HJ-Si/TiO₂/SS-83 in alkaline seawater under simulated AM 1.5 G illumination without iR compensation.** (a) J–V curves and (b) the corresponding ABPE curves. (c) Faradaic efficiency and production rate of H₂ and O₂ under the potential 1.5 V versus RHE of SS-83 and (d) the stability tests in alkaline natural seawater. (e) Summary of the reported H₂ production rate and stability of photoanodes applied in PEC seawater splitting[10,15–17,50–54]. The data in (a–e) were measured in alkaline natural seawater, simulated seawater and 1 M KOH plus 0.5 M NaCl electrolytes under simulated AM 1.5 G illumination.

structural reconstruction and deactivation[41]. As a result, the reaction environment on the surface of catalysts might further deteriorate due to the synergy of various halide ions, which results in an apparent decrease in stability. To simplify the complicated salinity in seawater, the influence from chloridion in seawater on semiconductor-based photoanode is investigated in this study, revealing the anticorrosion mechanism from the reconstructed stainless steel on seawater splitting. But it is of great significance to investigate the corrosion behaviors of photoelectrode in relation to all ions on seawater splitting in future research.

## Discussion

In summary, this paper describes the design and fabrication of a Si photoanode coated with a TiO₂/SS bilayer stack as protective and catalytic layer for stable PEC seawater splitting. The TiO₂ protective layer prevents the ITO layer of HJ-Si from defects induced by plasma damage during sputtering. Moreover, the chromium-incorporated NiFe (oxy)hydroxide reconstructed from stainless steel is an effective protective cocatalyst for seawater splitting with substantial OER activity enhancement as well as inhibiting the adsorption of Cl⁻. The fabricated photoanode achieves stable water splitting in chloride-containing alkaline electrolytes for 167 h. Furthermore, although the photoanode in simulated seawater shows decreased stability, this PEC system remains stable for 55 h with a recorded H₂ production rate (600 μmol h⁻¹ cm⁻²), a performance that stands out among existing seawater-based PEC systems for solar H₂ production. This design strategy opens a new pathway to accommodate the incompatible anticorrosion properties with reduced interfacial damages for low-cost stainless steel applied in semiconductor photoelectrodes, which may enable stable H₂ production from solar energy using abundant seawater.

## Methods

### Fabrication of Si electrode and the deposition of TiO₂

n-type c-Si wafers (1-3 Ω·cm resistivity, 155 μm thickness, and (100) orientation) were used as the starting substrate. The c-Si substrates were etched by a mixture of 5 wt% potassium hydroxide (KOH, Macklin Co., Ltd, 0.95) and 10 wt% isopropyl alcohol (IPA, tianjin real&lead Co., Ltd) for 20 min under 80 °C in order to produce the pyramid texture. After cleaning by the RCA method (NH₄OH/H₂O₂ aqueous solution, 80 °C), followed by HF etching (5% HF aqueous solution, tianjin kermel Co., Ltd, 60 s) to remove the native oxide, intrinsic a-Si layers (5 nm) were deposited by plasma-enhanced chemical vapor deposition (PECVD) at 200 °C. Then, highly doped n⁺-a-Si and p⁺-a-Si layers were also prepared by putting into PECVD chamber and depositing at 200 °C. n⁺-a-Si (5 nm) and p⁺-a-Si (5 nm) were fabricated next to intrinsic a-Si to form the heterojunction to separate photogenerated carriers. p-type a-Si was deposited using a gas mixture containing 4% silane diluted in hydrogen and BF₃ at 0.9 Torr and 40 W RF power for 200 s. n-type a-Si layer was deposited at 1.0 Torr, and 40 W RF power for 200 s, using a gas mixture containing 6.7% silane diluted in hydrogen with the PH₃:SiH₄ doping gas[42]. A layer of ITO with 80 nm thickness was prepared on both side of Si substrate by radio frequency (RF) magnetron sputtering (0.2 Pa, 40 W, 4 min, in 20 sccm Ar without heating), to increase the electrical conductivity of the surfaces while reduce the light reflection. The protective layer TiO₂ (10 nm) was prepared by ALD at 150 °C with 300 cycles. The thickness of TiO₂ is determined by several factors, including chemical stability, electrical conductivity, and film uniformity. Considering thicker TiO₂ might sacrifice the electrical conductivity, a thickness of 10 nm is selected for the TiO₂ layer, as TiO₂ exceeding 8 nm already demonstrates enough protection against plasma damage towards interfaces according to previous studies[27]. The cycle sequences are as

follows: titanium (IV) isopropoxide (TTIP, Sigma-Aldrich, ≥99.9999%) dosed for 3 s with $N_2$ purging for 8 s later; ultrapure water dosed for 0.2 s with $N_2$ purging for 8 s.

## The sputter of Ni and Pt catalytic layer

Stainless steel and Ni were deposited respectively through custom-made direct current (DC) magnetron sputtering. AISI 316 L and Ni targets were provided by Zhongnuo Advanced Material (Beijing) Technology Co., Ltd. To remove the residual gas, the base pressure of the chamber was maintained at $10^{-5}$ Pa, and then high purity Ar flow with 20 sccm gas flow was transported into the system. Finally, the chamber pressure was fixed at 1 Pa. The sputter power was kept at 10 W and sputter durations were adjusted.

## Preparation of Alkaline Seawater

Substitute seawater was prepared according to Standard Practice for the Preparation of Substitute Ocean Water (An American National Standard, Designation: D 1141–98 (2003)). To prepare the substitute ocean water, 24.534 g sodium chloride (NaCl, Heowns Co., Ltd, 0.99) and 40.94 g anhydrous sodium sulfate ($Na_2SO_4$, tianjin real&lead Co., Ltd) were dissolved in 800 mL water. Then Stock Solution No.1 (20 mL) and Stock Solution No. 2 (10 mL) were added with vigorous stirring. Finally, the solution was diluted to 1.0 L.

The compositions for preparing Stock Solution No.1 and Stock Solution No. 2 were shown as follows: Stock Solution No.1: $MgCl_2 \cdot 6H_2O$ (Tianjin fengchuan Co., Ltd, 55.56 g), $CaCl_2$(anhydrous) (China National Pharmaceutical Group Co., Ltd, 5.79 g) and $SrCl_2 \cdot 6H_2O$ (Tianjin fuchen Co., Ltd, 0.21 g) were dissolved in 100 mL water; Stock Solution No.2: KCl (Heowns Co., Ltd,, 6.95 g), $NaHCO_3$ (Tianjin fengchuan Co., Ltd, 2.01 g/L), KBr (Energy chemical Co., Ltd,, 1.0 g), $H_3BO_3$ (Aladdin Co., Ltd, 0.27 g/L) and NaF (Aladdin Co., Ltd, 0.03 g/L) were dissolved in 100 mL water.

The natural seawater was collected from BoHai Sea in China and filtered to remove sediment and other solids.

The preparation of alkaline seawater was shown as follows:

56.106 g of KOH was added to the 1 L seawater. Then the solution was stranded for 6 hours and filtered to prepare alkaline seawater.

## PEC measurements

Three-electrode configuration was fabricated for PEC measurements which consisted of the reference electrode (Hg/HgO, Gaoss Union, Co., Ltd), the counter electrode (platinum foil), and the prepared electrode. The potentials measured with Hg/HgO were converted to reversible hydrogen electrode potential using the Nernst equation:

$$E_{RHE} = E_{Hg/HgO} + 0.059pH + 0.098 \quad (1)$$

J-V curves were recorded with a scan rate of 20 mV s$^{-1}$ under AM 1.5 G solar simulator by an electrochemical workstation without iR-compensation at indoor temperature (25 °C). An AAA-class solar simulator (San-EI Electric Co., Ltd., Japan) equipped with a 150 W Xenon lamp and an AM 1.5 filter was used. Before the experiments, a calibrated Si photodiode[43,44] (Thorlabs, FDS100-CAL) was used to calibrate the power intensity of light to 100 mW/cm². Chronoamperometry was measured in the same system for a long-term stability test. Software Image J was used to measure the active geometric areas of back-illuminated electrodes.

The ABPEs were tested under the three-electrode configuration using the following equation, in which J (mA/cm²) is the photocurrent density from J-V curves, $V_b$ (vs. RHE) is the corresponding applied bias between the working electrode and the reference electrode, and P represented the incident illumination intensity with a value of

100 mW cm$^{-2}$

$$ABPE = J \times \frac{1.23 - |V_b|}{P} \times 100\% \quad (2)$$

The amount of $O_2$ was measured from gas chromatography (GC, Agilent 7890B). The amount of ClO$^-$ was measured according to N, N-diethyl-p-phenylenediamine sulfate (DPD) spectrophotometric method. The Faradaic efficiency of photoanode products was calculated as follows:

$$\text{Faraday Efficiency} = \frac{\text{Moles products} \times \text{Number of transferred electrons} \times \text{Faraday constant}}{\text{The number of electrons passed}} \times 100\% \quad (3)$$

The IPCE measuring system (Beijing Zolix, Solar Cell Scan 100) consists of Xenon lamp and monochromator light. Before measurement, the monochromatic light intensity of the system is corrected with a standard silicon solar cell provided with the IPEC system. The IPCE efficiency calculation formula is shown in the formula, where λ represents the incident light wavelength, $I_{light}$ represents the photocurrent density under illumination, $I_{dark}$ represents the current density under dark field conditions, and P represents the light intensity under the corresponding wavelength[43].

$$IPCE = \frac{\left[\frac{1240}{\lambda} \times (I_{light} - I_{dark})\right]}{P} \times 100\% \quad (4)$$

The ingetration of photocurrent based on the IPCE over standard AM 1.5 G spectrum (ASTM G173-03) is shown as follows[44]:

$$I_{Integrated} = \int_{\lambda=300}^{\lambda} q/hc \, IPCE(\lambda) \cdot E(\lambda) \cdot \lambda \cdot d\lambda \quad (5)$$

Where E is the irradiance in $W \cdot m^{-2} nm^{-1}$, h is the Planck's constant, c is the light speed.

## Characterization

The morphology was captured by field emission scanning electron microscope (FESEM, Siachi Regulus 8100) and element mapping was scanned by its energy dispersive spectroscopy. The Grazing incidence X-ray diffraction (GIXRD) (Smartlab) was characterized over a 2θ range from 20° to 80° at a scanning speed of 2° per step with Cu Kα radiation at 60 kV and 220 mA. The X-ray photoelectron spectroscopy was collected from ESCALAB Xi+ (ThermoFisher Scientific) with an Al Kα X-ray source (1486.6 eV) and the data were calibrated against the C 1 s photoelectron peak as the reference where binding energy located at 284.8 eV. The optical transmission spectra of the samples were measured by Shimadzu UV-3600 spectrophotometer. The thickness of $TiO_2$ and sputtered films were measured and fitted through spectroscopic ellipsometer (M-2000 DI, J. A. Woollam Co., Inc.).

## DFT calculations

Density functional theory (DFT) calculations were carried out via Vienna Ab Initio Simulation software (VASP)[45]. The electron exchange and correlation effects were described by the Perdew-Burke-Ernzerhof functional (PBE) form of the generalized gradient approximation (GGA)[46]. Considering van der Waals correction for all systems, the DFT-D3 method with Becke-Jonson damping was used[47]. The projector-augmented wave (PAW) method was used to describe the interaction between atomic cores and electrons[48]. The plane-wave basis set was employed with a cutoff energy of 400 eV. Meanwhile, the atomic force convergence criterion of force was set to 0.02 eV/Å. The 3-layer

surface slab models were chosen for $Ni(OH)_2$ and $Ni_{1-x-y}Fe_xCr_y(OH)_2$. The bottom layer of the above slabs was fixed at bulk structures, and the two top layers and the adsorbates were fully relaxed. A vacuum space with at least 15 Å was placed in the Z direction. We carefully tested the K-point mesh, and a 3×3×1 k-point grid was selected for all slabs. DFT + U method was employed to better describe the localized 3d electrons of Ni, Fe, and Cr in $Ni(OH)_2$ and $Ni_{1-x-y}Fe_xCr_y(OH)_2$, wherein $U_{Ni}$-$J_{Ni}$ = 6.4 eV, $U_{Fe}$-$J_{Fe}$ = 3.9 eV, and $U_{Cr}$-$J_{Cr}$ = 3.5 eV was adopted[49].

The OER reaction follows the 4-electron transfer process:

$$* + OH^- = *OH + e^- \tag{6}$$

$$*OH + OH^- = *O + H_2O + e^- \tag{7}$$

$$*O + OH^- = *OOH + e^- \tag{8}$$

$$*OOH + OH^- = O_2 + * + H_2O + e^- \tag{9}$$

The OER performances were evaluated by calculating the reaction free energy of each step:

$$\Delta G1 = G(*OH) - G(*) + \frac{1}{2}G(H_2) - G(H_2O) - eU - kTln\left(\frac{1}{[H^+]}\right)$$
$$= G(*OH) - G(*) + \frac{1}{2}G(H_2) - G(H_2O) - eU\text{-}0.0592pH \tag{10}$$

$$\Delta G2 = G(*O) - G(*OH) + \frac{1}{2}G(H_2) - eU - kTln\left(\frac{1}{[H^+]}\right)$$
$$= G(*O) - G(*OH) + \frac{1}{2}G(H_2) - eUrhe \tag{11}$$

$$\Delta G3 = G(*OOH) - G(*O) + \frac{1}{2}G(H_2) - G(H_2O) - eU - kTln\left(\frac{1}{[H^+]}\right)$$
$$= G(*OOH) - G(*O) + \frac{1}{2}G(H_2) - G(H_2O) - eU\text{-}0.0592pH \tag{12}$$

$$\Delta G4 = G(O_2) + G(*) + \frac{1}{2}G(H_2) - G(*OOH) - eU - kTln\left(\frac{1}{[H^+]}\right)$$
$$= G(O_2) + G(*) + \frac{1}{2}G(H_2) - G(*OOH) - eU\text{-}0.0592pH \tag{13}$$

where G(*OH), G(*O) and G(*OOH) represent the Gibbs free energy of the intermediates (*OH, *O and *OOH) that is adsorbed on active sites. G(*) is the Gibbs free energy of a clean slab without any intermediate. $G(H_2)$, $G(O_2)$ and $G(H_2O)$ represent the Gibbs free energy of gas $H_2$, $O_2$ and liquid $H_2O$. The calculation was based on a computational standard hydrogen electrode model.

Metal chloride-hydroxide corrosion mechanisms:

$$M + Cl^- = MCl_{ads} + e^- \tag{14}$$

$$MCl_{ads} + Cl^- = MCl_X^- + e^- \tag{15}$$

$$MCl_X^- + OH^- = M(OH)_X + e^- \tag{16}$$

In particular, we defined ΔG to show the adsorption difference of the intermediate on doped $Ni_{1-x-y}Fe_xCr_y(OH)_2$ and initial $Ni(OH)_2$.

$$\Delta G(*Cl) = G_{Ni_{1-x-y}Fe_xCr_y(OH)_2}(*Cl) - G_{Ni(OH)_2}(*Cl) \tag{17}$$

where $G_{Ni_{1-x-y}Fe_xCr_y(OH)_2}(*Cl)$ is the Gibbs free energy of the intermediate on $Ni_{1-x-y}Fe_xCr_y(OH)_2$, $G_{Ni(OH)_2}(*Cl)$ is the Gibbs free energy of the intermediate on initial $Ni(OH)_2$.

## Data availability

All data generated or analysed during this study are included in the published article and its Supplementary Information. Source data are provided with this paper.

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

## Acknowledgements

We acknowledge the National Key R&D Program of China (2021YFA1500804), the National Natural Science Foundation of China (22121004, 22250008, 51861125104, and 22038009), the Program of Introducing Talents of Discipline to Universities (No. BP0618007) and the Xplorer Prize for financial support.

## Author contributions

J.L.G and T.W. supervised the project. J.L.G., T.W. and S.X.Z. conceptualized the project. S.X.Z. and B.L. developed the concept and carried out the experiments and data analysis. Catalyst synthesis procedures were developed and performed by G.Z. and S.J.W. Theoretical calculations were carried out by Z.J.Z. and K.L.L. All authors discussed the results and participated in writing of the manuscript.

## Competing interests

There are no conflicts to declare.
