## [Peer Review File · Nature Communications]

A silicon photoanode protected with TiO₂/stainless steel bilayer stack for solar seawater splittingREVIEWER COMMENTS

Reviewer #1 (Remarks to the Author):

after the modifications introduced by the authors, the manuscript has been improved.

Now it can be considered as candidate to the published.

Nevertheless, in spite of the efforts performed, it should be taking into account that manuscript is offering solely a variation about how to prepare the electrodes TiO₂/SS but it is not offering any new advantages concerning solar to hydrogen conversion, lifetime, degradation, scalability, stability,.....and competitiveness with other technologies.

Reviewer #2 (Remarks to the Author):

This study presents a novel approach to engineering a silicon photoanode by employing a dual-purpose bilayer coating of titanium dioxide and stainless steel. This coating not only serves as a protective layer but also acts as a catalytic agent, significantly enhancing the photoanode's efficiency in photoelectrochemical seawater splitting. The research presents a notable advancement in integrating anti-corrosion features and minimizing interfacial damage, especially relevant when using low-cost stainless steel in semiconductor photoelectrodes. The findings hold substantial promise, potentially leading to stable hydrogen production from solar energy using seawater, an abundant resource.

However, the manuscript requires addressing the following concerns before consideration for acceptance:

On page 3, line 54, there's a typographical error: "AD-DC" should be corrected to "AC-DC".

While the authors report performance surpassing previous studies, there is a lack of detailed information on measurement conditions. It's essential to provide the specifications of the solar simulator used, including its model number and the class of solar simulation quality (e.g. AAA-class). Additionally, a comparison between the Internal Quantum Efficiency (IQE) of the calibration device and the Incident Photon-to-Current Efficiency (IPCE) of the record efficiency device must be presented to determine if any corrections are needed.

The authors suggest that the TiO₂ layer protects against sputtering damage to the semiconductor layer which degrades its minority carrier transport. However, since the silicon device is already covered with a thick Indium Tin Oxide (ITO) layer, and typical sputtering damage is thinner than this, the damage to the Si layer seems negligible in this context. More supporting data is required to substantiate this claim convincingly.

As previously noted by another reviewer, the details regarding device fabrication are insufficient.

Additional information on the p+ and n+ amorphous silicon layers, including the dopants and deposition conditions, is needed.

Reviewer #3 (Remarks to the Author):

This paper describes the fabrication of a bilayer stack of stainless steel and TiO₂ as a cocatalyst and protective layer for Si photoanode, which enhances OER activity and inhibits Cl⁻ adsorption. Chromium-infused NiFe(oxy)hydroxide from the stainless steel film shields the Si substrate from seawater corrosion and promotes oxygen generation, while TiO₂ protects the Si substrate from plasma damage. The fabricated photoanode achieves stable water splitting in chloride-containing alkaline electrolytes for 167 h, while remaining stable for 39 h with a recorded H₂ production rate (590 μmol h⁻¹ cm⁻²). The authors have improved the manuscript and properly addressed the reviewers' comments. This manuscript is strongly recommended for publication in Nature Communications after addressing the following minor issues:

1. The authors may need to discuss the existing and competing approaches to effectively utilize seawater, such as dual reverse osmosis systems. A discussion and comparison must be done.
2. More details are needed to analyze the interfaces such as silicon/TiO₂ and TiO₂/SS. Why do the authors use only 10 nm TiO₂ protective layer? What is the evolution of the TiO₂ after that? What is the structure of the deposited TiO₂?
3. The EIS spectra should be accompanied by an equivalent circuit diagram.
4. It is mentioned that the design and fabrication of various sputtered stainless steel films deposited on the silicon photoanode completely isolates the electrolytes and the semiconductor substrate. What happened to the surface of the photoanode after the stability test?
5. The authors should explain why SS-83 shows the best PEC seawater splitting performance.
6. More details on the cross section of the bilayer stack should be provided. A detailed analysis of the reconstruction of the sputtered SS layer from the outer interface with the electrolyte to the inner interface with the TiO₂ should be provided.
7. The authors should check their manuscript carefully to avoid errors. For example, the text in Figure S7 has a typo; in Figure 2b, "HADDF" should be "HAADF".

Reviewer #1

General Comment R1: *After the modifications introduced by the authors, the manuscript has been improved. Now it can be considered a candidate to the published.*

Response: We thank the reviewer for the valuable comments.

Specific Comment R1-1: *Nevertheless, in spite of the efforts performed, it should be taken into account that the manuscript is offering solely a variation about how to prepare the electrodes TiO₂/SS but it is not offering any new advantages concerning solar to hydrogen conversion, lifetime, degradation, scalability, stability, and competitiveness with other technologies.*

Response: We thank the reviewer for the critical and detailed comments. We fully understand the review's concerns on the mentioned factors and the related explains are as follows:

It may not be that straightforward to obtain the solar to hydrogen conversion (STH) value of the proposed system because this work just focused on the design of the photoanode. For the photoanode, an ABPE of 2.62% could be obtained in alkaline natural seawater, while other photoanodes applied in PEC seawater splitting systems illustrate ABPE less than 1% (Supplementary Table 10), indicating the remarkable energy conversion efficiency of HJ-Si/TiO₂/SS. In addition, we also predict the hydrogen production rate of various PEC seawater splitting systems, compared with HJ-Si/TiO₂/SS measured through GC, indicating that hydrogen production of this work is ten times more efficient than other PEC seawater splitting systems.

The stabilities of photoanode coated with bilayer stack TiO₂/SS were evaluated in different circumstances, including 1 M KOH + 0.5 M NaCl, alkaline natural seawater, and the day-night cycling tests. The fabricated photoanode achieves stable water splitting in alkaline natural seawater for 55 h. However, other materials for photoanode suffer from severe corrosion in seawater without a protective layer (Supplementary Table 10), such as BiVO₄, WO₃, TiO₂, etc. Therefore, the photoanodes in this study illustrate excellent stability, compared with recent reports on various PEC seawater splitting systems (Supplementary Table 10).

Commercial-scale solar seawater splitting devices have been considered to be cost-competitive with fossil-based fuels. The scalability of photoelectrodes faces more intractable challenges, including depositing uniform protective films with large areas and industrial electrolyzers coupled well with photoelectrodes. Some researchers have elucidated how flow patterns and heat transfer may influence the deposition of protective layers over large photoelectrodes, guiding future industrial applications of PEC water splitting (*ACS Appl. Mater. Interfaces* 2023, 15, 1138–1147).

Currently, the main routes to effectively utilize seawater include indirect and direct seawater splitting. Direct seawater splitting contains the photocatalysis system and PEC system. Although some researchers recently investigated that indium gallium nitride

photocatalyst achieved an STH efficiency of 6.2% in a large-scale photocatalytic water-splitting system with a natural solar light capacity of 257 watts (*Nature*, 2023, 613, 66), this technology also faces several challenges, such as gas separation of H₂ and O₂ and precious metal photocatalyst as well as the cost of indium and gallium elements. Therefore, based on the Si substrate and mature thin film process, the photoanode could be fabricated in an efficient and economical method and effectively combined with the mature electrolyzer with a light window, avoiding the risk of product mixing.

Although whether direct seawater splitting is the preferable method is still under debate, the design of Si photoanode protected with TiO₂/stainless steel bilayer enables the direct utilization of solar energy, without having to build an additional micro-grid that is necessary for photovoltaic or wind turbine-powered electrolyzers, providing alternative solutions to the scenarios where the space is limited such as ships or offshore platforms.

We have revised the content in the revised manuscript Paragraph 3 Page 13 “The hydrogen production rate of various PEC seawater splitting systems is predicted, compared with HJ-Si/TiO₂/SS measured through GC, indicating that hydrogen production of Si photoanode with the bilayer stack is ten times more efficient than other PEC seawater splitting systems, which stands out among previously reported photoanodes on seawater splitting.”

Paragraph 1 Page 3 “In terms of photocatalysis seawater splitting, although some researchers recently investigated that indium gallium nitride photocatalyst achieved solar to hydrogen efficiency of 6.2% in a large-scale photocatalytic water-splitting system, this technology also faces several challenges, such as gas separation of H₂ and O₂ and precious metal photocatalyst.”

Reviewer #2

General Comment R2: *This study presents a novel approach to engineering a silicon photoanode by employing a dual-purpose bilayer coating of titanium dioxide and stainless steel. This coating not only serves as a protective layer but also acts as a catalytic agent, significantly enhancing the photoanode's efficiency in photoelectrochemical seawater splitting. The research presents a notable advancement in integrating anti-corrosion features and minimizing interfacial damage, especially relevant when using low-cost stainless steel in semiconductor photoelectrodes. The findings hold substantial promise, potentially leading to stable hydrogen production from solar energy using seawater, an abundant resource. However, the manuscript requires addressing the following concerns before consideration for acceptance:*

Response: We thank the reviewer for the valuable comments. We have revised the manuscript carefully with additional experimental results according to the Specific Comment.

Specific Comment R2-1: *On page 3, line 54, there's a typographical error: "AD-DC" should be corrected to "AC-DC".*

Response: We acknowledge the helpful suggestion of the reviewer. We have revised the content in the revised manuscript Paragraph 1 Page 3 “**Compared with direct seawater electrolysis powered by photovoltaics powered electrolyzers, PEC seawater splitting could utilize solar energy directly without additional micro-grids or AC-DC devices, which may further simplify the process of H₂ production from seawater.**”

Specific Comment R2-2: *While the authors report performance surpassing previous studies, there is a lack of detailed information on measurement conditions. It's essential to provide the specifications of the solar simulator used, including its model number and the class of solar simulation quality (e.g. AAA-class). Additionally, a comparison between the Internal Quantum Efficiency (IQE) of the calibration device and the Incident Photon-to-Current Efficiency (IPCE) of the record efficiency device must be presented to determine if any corrections are needed.*

Response: We thank the reviewer for the precious comment and the suggestion. An AAA-class solar simulator (San-EI Electric Co., Ltd., Japan) equipped with a 150 W Xenon lamp and an AM 1.5 filter was used. Before the experiments, a calibrated Si photodiode (FDS100-CAL, Thorlabs) was used to calibrate the power intensity of light to 100 mW/cm². All the experiments were conducted under simulated AM 1.5G illumination with 100 mW/cm². The spectrum of the A 150 W Xenon lamp (San-EI Electric Co., Ltd., Japan) and the drawing of quartz photoelectrochemical cells is provided as follows:

(Revised) Supplementary Fig. 9 (a) The digital photograph and (b) the schematic diagram of the PEC cell. (c) The specific position of three-electrode on the lid. (d) The spectrum of the A 150 W Xenon lamp with AM 1.5G filter.

We totally agree that comparison between IQE and IPCE of the calibration device would be useful to understand the advantages of Si photoanode protected with TiO₂/stainless steel bilayer stack. Unfortunately, our IPCE and IQE instrument is out of order. We also actively seek to use the IPCE instruments from other groups in our university but their equipment is also in failure because this type of instrument is not used much in our recent research. However, we have provided additional data of several factors, such as current density at 1.23 V, ABPE, stable duration, and hydrogen production rate of the reported PEC seawater splitting system, thus achieving reasonable and well-rounded comparison of photoanodes.

We have revised related description of solar simulator in Method Page 18 “An AAA-class solar simulator (San-EI Electric Co., Ltd., Japan) equipped with a 150 W Xenon lamp and an AM 1.5 filter was used.” And we also added the spectrum of the A 150 W Xenon lamp with AM 1.5G filter in the revised Supplementary Fig. 9.

Specific Comment R2-3: *The authors suggest that the TiO₂ layer protects against sputtering damage to the semiconductor layer which degrades its minority carrier transport. However, since the silicon device is already covered with a thick Indium Tin Oxide (ITO) layer, and typical sputtering damage is thinner than this, the damage to the Si layer seems negligible in this context. More supporting data is required to substantiate this claim convincingly.*

Response: We acknowledge the helpful suggestion of the reviewer. We apologize that we presented a misleading description in the original submission. The Si substrate of HJ structure is covered with an ITO layer, approximately 80 nm, which can be seen in

the high-resolution cross-sectional TEM. The ITO surface also suffers from defects formation during the sputtering process. Therefore, The TiO₂ protective layer can prevent the ITO layer from plasma damage during sputtering, resulting in improved PEC performance.

As suggested by the reviewer, we revised the related description in the Abstract “while the TiO₂ is capable of avoiding the plasma damage of the surface layer of Si photoanode during the sputtering of stainless steel catalysts”

In Paragraph 3 Page 5 “The Si photoanode (Supplementary Fig. 1) was fabricated by depositing TiO₂ and stainless steel films on a heterojunction Si substrate (ITO/n⁺-a-Si/a-Si/c-n-Si/a-Si/p⁺-a-Si/ITO, denoted as HJ-Si).” “The improved PEC performance could be attributed to the fact that TiO₂ effectively passivates the defects on the ITO layer of HJ-Si induced from the plasma damage as well as the Fermi level pinning at the metal/semiconductor interface²⁷.”

In Page 16 Discussion “The TiO₂ protective layer prevents the ITO layer of HJ-Si from defects induced by plasma damage during sputtering”

In Page 17 Method “A layer of ITO with 80 nm thickness was prepared on both side of Si substrate by radio frequency (RF) magnetron sputtering (0.2 Pa, 40 W, 4 min, in 20 sccm Ar without heating), to increase the electrical conductivity of the surfaces while reduce the light reflection.”

Specific Comment R2-4: *As previously noted by another reviewer, the details regarding device fabrication are insufficient. Additional information on the p⁺ and n⁺ amorphous silicon layers, including the dopants and deposition conditions, is needed.*

Response: We apologize for not giving abundant information of device fabrication. As suggested by the reviewer, we added the details of preparation in the manuscript Method Page 17 “Intrinsic a-Si layers (5 nm) were deposited by plasma-enhanced chemical vapor deposition (PECVD) at 200 °C. Then, highly doped n⁺-a-Si and p⁺-a-Si layers were also prepared by putting into PECVD chamber and depositing at 200 °C. n⁺-a-Si (5 nm) and p⁺-a-Si (5 nm) were fabricated next to intrinsic a-Si to form the heterojunction to separate photogenerated carriers. A layer of ITO with 80 nm thickness was prepared on both side of Si substrate by radio frequency (RF) magnetron sputtering (0.2 Pa, 40 W, 4 min, in 20 sccm Ar without heating), to increase the electrical conductivity of the surfaces while reduce the light reflection. The protective layer TiO₂ (10 nm) was prepared by ALD at 150 °C with 300 cycles. The thickness of TiO₂ is determined by several factors, including chemical stability, electrical conductivity, and film uniformity. Considering thicker TiO₂ might sacrifice the electrical conductivity, a thickness of 10 nm is selected for the TiO₂ layer, as TiO₂ exceeding 8 nm already demonstrates enough protection against plasma damage towards interfaces according to previous studies²⁷”

Reviewer #3

General Comment R3: *This paper describes the fabrication of a bilayer stack of stainless steel and TiO₂ as a cocatalyst and protective layer for Si photoanode, which enhances OER activity and inhibits Cl⁻ adsorption. Chromium-infused NiFe(oxy)hydroxide from the stainless steel film shields the Si substrate from seawater corrosion and promotes oxygen generation, while TiO₂ protects the Si substrate from plasma damage. The fabricated photoanode achieves stable water splitting in chloride-containing alkaline electrolytes for 167 h, while remaining stable for 39 h with a recorded H₂ production rate (590 μmol h⁻¹ cm⁻²). The authors have improved the manuscript and properly addressed the reviewers' comments. This manuscript is strongly recommended for publication in Nature Communications after addressing the following minor issues:*

Response: We thank the reviewer for the valuable comments. We have revised the manuscript carefully with additional experimental results according to the Specific Comment.

Specific Comment R3-1: *The authors may need to discuss the existing and competing approaches to effectively utilize seawater, such as dual reverse osmosis systems. A discussion and comparison must be done.*

Response: We thank the reviewer very much for the precious comment about comparing different pathways in seawater splitting. Compared with water electrolysis, the utilization of seawater into hydrogen is still under insensitive development, mostly due to intractable side reactions and corrosion from complex ions in seawater. Currently, the main routes to make hydrogen from seawater include indirect and direct seawater splitting. Some previous research (*Energy Environ. Sci.* 2021, 14, 4831) suggest that the indirect seawater splitting, e.g., traditional water electrolysis coupled with seawater reverse osmosis (SWRO), presents economic superiority to direct seawater splitting because of mature desalination and deionization. Some researchers (*Energy Environ. Sci.* 2021, 11, 3679) also analyzed the thermodynamic requirements, energy consumption, and capital costs of devices of the two ways, indicating that double reverse osmosis systems coupled with conventional electrolyzer are promising.

However, conventional electrolyzers are susceptible to dissolved salts in seawater. Although the seawater is desalinated, electrolyte fed into conventional electrolyzers should be reduced to less than 10 ppm (*Joule*, 2023, 7, 20; *Nat. Energy*, 2020, 5, 367). In addition, it is necessary to remove the organic molecules and dissolved gases in advance in order to produce high-purity hydrogen and lengthen the electrolyzer's lifetime (*Nature*, 2022, 612, 673). But these additional steps will increase the expenditure and the complexity of devices. Thus, direct seawater splitting is considered a remarkable method to utilize seawater for H₂ production.

Although whether direct seawater splitting is the preferable method is still under debate, the design of Si photoanode protected with TiO₂/stainless steel bilayer enables

the direct utilization of solar energy, without having to build an additional micro-grid that is necessary for photovoltaic or wind turbine-powered electrolyzers, providing alternative solutions to the scenarios where the space is limited such as ships or offshore platforms.

To make this point clear to readers, we have revised and added a related descriptions in the revised manuscript Paragraph 1 Page 3 “Currently, the main routes to produce hydrogen from seawater include indirect and direct seawater splitting. Although indirect seawater splitting, such as traditional water electrolysis coupled with seawater reverse osmosis (SWRO), integrates several mature technology, it also increases the expenditure and the complexity of the system.”

Specific Comment R3-2: *More details are needed to analyze the interfaces such as silicon/TiO₂ and TiO₂/SS. Why do the authors use only 10 nm TiO₂ protective layer? What is the evolution of the TiO₂ after that? What is the structure of the deposited TiO₂?*

Response: We thank the reviewer very much for the valuable comments. TiO₂ thin film deposited by ALD presents several advantages of chemical stability, high conductivity, and high uniformity, which is widely used in PEC water splitting and achieves numerous remarkable results (Science, 2014, 344, 1005; Energy Environ. Sci. 2015, 8, 650). It is confirmed that the thickness of TiO₂ presents no significant effect on the stability of Si photoanode when its thickness exceeds 8 nm, while thicker TiO₂ might pose a threat to the conductivity of photoelectrode. Thus, taking into account all factors, 10 nm TiO₂ protective layer is selected, which is based on our previous work (Energy Environ. Sci. 2020, 13, 221; Adv. Funct. Mater. 2021, 31, 2007222).

The cross-sectional HRTEM of pristine and activated samples implies that the structure of TiO₂ is amorphous which shows no obvious transformation during activation. The protective layer TiO₂ (10 nm) was prepared by ALD at 150 °C with 300 cycles (Nat. Commun. 2022, 13, 7111).

(Revised) Fig. 2. High-resolution cross-sectional TEM of (c) pristine and (d) activated HJ-Si/TiO₂/SS.

To make this point clear to readers, we have revised and added related descriptions in the revised manuscript Paragraph 3 Page 5 “The thickness of TiO₂ presents no significant effect on the stability of Si photoanode when its thickness exceeds 8 nm, while thicker TiO₂ might pose a threat to the conductivity of photoelectrode. Thus, considering both the conductivity and protection, the TiO₂ layer with 10 nm is selected.”

In Page 17 Method“The thickness of TiO₂ is determined by several factors, including chemical stability, electrical conductivity, and film uniformity. Considering thicker TiO₂ might sacrifice the electrical conductivity, a thickness of 10 nm is selected for the TiO₂ layer, as TiO₂ exceeding 8 nm already demonstrates enough protection against plasma damage towards interfaces according to previous studies²⁷”

Specific Comment R3-3: *The EIS spectra should be accompanied by an equivalent circuit diagram.*

Response: We thank the reviewer for the critical comment. As suggested by the reviewer, electrochemical impedance spectroscopy is accompanied by the equivalent circuit diagram to investigate the charge carrier transport of photoanodes. The Nyquist impedance plots (Supplementary Fig.9) have been added for representative samples measured under illumination at 1.23 V vs RHE. It is obvious that the presence of SS with different thicknesses reduces the carrier transfer resistances in the solid/electrolyte interface, which indicates the excellent OER activity of the SS films, consistent with the remarkable onset potential and ABPE.

We have added the equivalent circuit diagram in revised Supplementary Fig.10.

(Revised) Supplementary Fig.10 The EIS of photoanodes coated with Ni-20, SS-20 and SS-8.

Specific Comment R3-4: *It is mentioned that the design and fabrication of various sputtered stainless steel films deposited on the silicon photoanode completely isolates the electrolytes and the semiconductor substrate. What happened to the surface of the photoanode after the stability test?*

Response: We are grateful to the reviewer for the valuable comment. We have provided the surface morphology and J-V characteristics of photoanodes after stability tests in different circumstances.

The PEC performance and structure of the photoanode show negligible changes after the stability test in 1 M KOH + 0.5 M NaCl (Fig. 4a and Supplementary Fig. 15), implying the superior anti-corrosion and catalytic ability of SS-83. In addition, upon another 168 h stability test with cyclic 8 h illumination followed by 16 h dark (open circuit condition) to mimic the day-night cycling operation, the surface of the photoanode after day-night cycling stability is also evaluated (Supplementary Fig. 17), illustrating the intact morphology.

Therefore, the slight change in surface morphology and PEC performance before and after stability tests illustrate the complete isolation between electrolytes and semiconductor substrates.

Specific Comment R3-5: *The authors should explain why SS-83 shows the best PEC seawater splitting performance.*

Response: We acknowledge the helpful suggestion of the reviewer. Compared with SS-20, SS-83 illustrates better OER activity and stability in 1 M KOH + 0.5 M NaCl because a thicker stainless steel cocatalyst totally isolates the semiconductor substrate from the corrosive ions in seawater. By contrast, there might be voids or pit holes in the thinner SS cocatalyst (20 nm), which results in the preferential adsorption of chlorides on the nonuniform regions of the cocatalyst, causing the structural collapse of SS-20. However, with the increasing thickness of SS cocatalysts, the PEC performance decreases rapidly, which might result from the increase in sheet resistance of SS-123 (Revised Supplementary Table 6), indicating that SS-83 efficiently reduces the carrier transfer resistances in the solid/electrolyte interface. Therefore, considering the sheet resistance and protective layer function of SS thin films, SS-83 is the best thickness for the photoanode.

As suggested by the reviewer, we have revised and added related descriptions in the revised manuscript Paragraph 3 Page 11 “**This could be attributed to the fact that thicker stainless steel cocatalyst totally isolates the semiconductor substrate from the corrosive ions in seawater. By contrast, there might be voids or pit holes in the thinner SS cocatalyst (20 nm), which results in the preferential adsorption of chlorides on the nonuniform regions.**”

Specific Comment R3-6: *More details on the cross-sectional of the bilayer stack should be provided. A detailed analysis of the reconstruction of the sputtered SS layer from the outer interface with the electrolyte to the inner interface with the TiO₂ should be provided.*

Response: We acknowledge the helpful suggestion of the reviewer. We have provided cross-sectional TEM images of photoanodes upon reconstruction, which could be used to understand the retarding mechanism of the bilayer stack.

Before reconstruction, the heterojunction Si substrate is covered by dense TiO₂ film and the pristine SS film before activation is compact without pin-holes or defects, completely isolated from the electrolyte-containing seawater. The structure of pristine SS film is nanocrystalline with discrete amorphous domains, which has been confirmed to improve pitting resistance in chloride-containing environments. (*Corros. Sci.* 2016, 104, 71).

After reconstruction, no pit holes or defects appear between TiO₂/SS and Si/ TiO₂ interfaces, which could be attributed to the nanocrystalline structure with discrete amorphous domains in the SS layer. In contrast, chloridion tends to erode among intergrain regions, resulting in high pit hole density on the materials with high crystallites. (*J. Am. Chem. Soc.* 2014, 136, 6191).

(Revised) Fig. 2. Morphologies and chemical states of HJ-Si/TiO₂/SS. High-resolution cross-sectional TEM of (c) pristine and (d) activated HJ-Si/TiO₂/SS.

As suggested by the reviewer, we have revised and added related descriptions in the revised manuscript Paragraph 2 Page 9 “Before reconstruction, the heterojunction Si substrate is covered by dense TiO₂ film and the pristine SS film before activation is compact without pin-holes or defects, while after reconstruction there are no pit holes in the SS layers as well as near the TiO₂/SS and Si/ TiO₂ interfaces, which could be attributed to the nanocrystalline structure with discrete amorphous domains in the SS layer.”

Specific Comment R3-7: The authors should check their manuscript carefully to avoid errors. For example, the text in Figure S7 has a typo; in Figure 2b, "HADDF" should be "HAADF".

Response: We appreciate the reviewer’s positive and constructive comments. We have carefully proofread the manuscript and corrected typo. We have revised the description in Supplementary Fig. 7 and Figure 2b.

(Revised) Supplementary Fig. 7 The high-resolution X-ray photoelectron spectroscopy of O 1s to HJ-Si/TiO₂/SS after activation.

(Revised) Fig. 2. Morphologies and chemical states of HJ-Si/TiO₂/SS.

REVIEWER COMMENTS

Reviewer #2 (Remarks to the Author):

While the authors have addressed some of the feedback, I believe that the manuscript still requires additional supporting data, especially to enhance the credibility of the recorded photocurrent on their device. Given that the manuscript presents a device with a record-high current density to substantiate their findings, it is imperative for the authors to adhere to the guidelines outlined in a Nature publication (<https://www.nature.com/articles/s41560-023-01432-3>).

1. Specifically, the authors must provide the Incident Photon-to-Current Efficiency (IPCE) of the device and present the integrated short-circuit current density (J_{sc}) derived from the IPCE spectrum. Furthermore, it is important to note that the device used for calibrating their solar cell does not follow the conventional method, which typically involves using a calibrated solar cell device with measured output data under standard 1-Sun conditions.
2. Figure 9(d) in the supplementary information appears to be sourced from the dataset provided by the manufacturer of their solar simulator, without any accompanying reference to verify its origin.
3. In the experimental section, it is advisable to include information about the dopants used and their respective doping levels.

Reviewer #3 (Remarks to the Author):

The authors have addressed all the comments and the manuscript can be accepted now.

Reviewer #2

General Comment R2: While the authors have addressed some of the feedback, I believe that the manuscript still requires additional supporting data, especially to enhance the credibility of the recorded photocurrent on their device. Given that the manuscript presents a device with a record-high current density to substantiate their findings, it is imperative for the authors to adhere to the guidelines outlined in a Nature publication (<https://www.nature.com/articles/s41560-023-01432-3>).

Response: We thank the reviewer for the valuable comments. We have revised the manuscript carefully with additional experimental results to follow the *Solar Cells Reporting Summary*.

Specific Comment R2-1: Specifically, the authors must provide the Incident Photon-to-Current Efficiency (IPCE) of the device and present the integrated short-circuit current density (J_{sc}) derived from the IPCE spectrum. Furthermore, it is important to note that the device used for calibrating their solar cell does not follow the conventional method, which typically involves using a calibrated solar cell device with measured output data under standard 1-Sun conditions.

Response: We acknowledge the helpful suggestion of the reviewer. We have added the Incident Photon-to-Current Efficiency (IPCE) of our device as a function of wavelength from 300 to 1100nm at the potential of 1.5 V vs. RHE. By integrating the measured IPCE over the standard AM 1.5G spectrum (ASTM G173-03), a photocurrent density of 32 mA/cm² could be calculated, which is very close to that of 34 mA/cm² at 1.5 V vs. RHE from the J-V test (Fig. 3a) under our AM 1.5G simulator. The consistency between the two sets of photocurrent density demonstrates the accuracy of our AM 1.5G simulator in simulating sunlight, ensuring the reliability and accuracy of J-V and IPCE measurements.

Supplementary Fig. 9 (d) The IPCE measured at 1.5 V vs. RHE with integrated current of photoanode against standard AM 1.5G spectrum (ASTM G173-03).

We thank the reviewer for the suggestions on calibrating solar cells. In our method, a calibrated Si photodiode (Thorlabs, FDS100-CAL) was used to calibrate the power

intensity of our AM 1.5G simulator to 100 mW/cm², while the light intensity of the IPCE system is corrected with a standard silicon solar cell provided with the instrument. Indeed, the calibrated Si photodiode shares the same mechanism with calibrated solar cell device. The calibrated Si photodiode (Thorlabs, FDS100-CAL) is featured by well-defined size and carefully measured output data under standard 1-Sun conditions, enabling an accurate calibration that has been adopted by many previous works (N. S. Lewis, Science, 2010, 327, 185-187 (SI) 10.1126/science.1180783; N. S. Lewis, Science, 2014, 344, 1005 (SI) 10.1126/science.1251428).

The description of Thorlabs FDS100-CAL can be found at:

https://www.thorlabs.com/newgrouppage9.cfm?objectgroup_id=2822;
<https://www.thorlabs.com/thorproduct.cfm?partnumber=FDS100-CAL>

The detailed specifications of our FDS100-CAL are scanned as following:

Thorlabs GmbH
 Muenchener Weg 1
 85232 Bergkirchen
 Germany
 Tel.: +49 8131 5956 0
 Fax.: +49 8131 5956 99

<http://www.thorlabs.com>

Certificate of Calibration No 22215073309

X New Unit
 Returned Unit

Result as left: PASS

Manufacturer Thorlabs GmbH
 Model Number FDS100-CAL
 Serial Number 220803101
 Date of Calibration 3-Aug-2022

NO	SCAN	MANUFACTURER	MODEL	SERIAL NO	CERTIFICATION	LAST CAL
1	CAL1	Hamamatsu	S2281	OK014	73340 21 P18	6-Jun-2021
2	CAL2	Thorlabs	GM10HS	TH0506	73327 21 P18	6-Dec-2021
3	CAL4	Thorlabs	S148C	15050530	NIST 36332	0-May-2022
4	CAL4	Thorlabs	S180C	190605400	NIST 36332	6-May-2022

Thorlabs GmbH does hereby certify that the above mentioned equipment has been calibrated in accordance with our quality management system.
 Our Quality Management System is certified according to DIN EN ISO 9001.

The measurement equipment used for calibration is traceable to national standards of the 'EUROMET' members (NPL, PTB, BNM etc.), the US 'NIST' or other national metrological institutions. Measurements which cannot be traced to national standards can be traced to natural constants, other accepted standards or relational measurements.

Additional documentation concerning traceability of the measurement equipment is available and can be examined upon request.
 The certificate of calibration may only be forwarded in complete form without any changes.

The recommended calibration interval is 12 months. The calibration period of this instrument / system begins on the date of receipt by the customer.

Calibrated by *A. Breitbart* Axel Breitbart
 Thorlabs GmbH

Date Received _____

Calibrated Due _____

Form 1005-01 Rev E Page 1 of 2

Test Report - as left

Model FDS100-CAL Temperature 24.5°C
 Serial No 220803101 Humidity 51% Test Date 3-Aug-2022
 Scan CAL1 Result Tester Axel Breitbart
 Result PASS

A [nm]	n [AWG]	A [nm]	n [AWG]	A [nm]	n [AWG]	A [nm]	n [AWG]	A [nm]	n [AWG]
350	4.88E-02	730	4.33E-01	1090	1.27E-01				
360	4.58E-02	730	4.43E-01	1100	9.92E-02				
380	4.48E-02	730	4.84E-01						
400	4.78E-02	730	4.84E-01						
420	5.48E-02	730	4.74E-01						
430	5.38E-02	730	4.84E-01						
440	7.44E-02	730	4.83E-01						
450	8.55E-02	730	4.83E-01						
460	9.70E-02	800	5.11E-01						
480	1.09E-01	810	5.20E-01						
490	1.20E-01	820	5.28E-01						
490	1.31E-01	830	5.36E-01						
490	1.43E-01	840	5.44E-01						
490	1.54E-01	850	5.52E-01						
500	1.78E-01	860	5.68E-01						
510	1.97E-01	870	5.77E-01						
520	2.11E-01	880	5.73E-01						
530	2.22E-01	910	5.84E-01						
540	2.34E-01	920	5.91E-01						
550	2.45E-01	930	6.00E-01						
570	2.57E-01	940	6.08E-01						
580	2.68E-01	950	6.16E-01						
590	2.79E-01	960	6.24E-01						
600	2.92E-01	970	6.32E-01						
620	3.18E-01	980	6.39E-01						
630	3.30E-01	1000	6.50E-01						
640	3.42E-01	1010	6.58E-01						
650	3.54E-01	1020	6.66E-01						
660	3.66E-01	1030	6.74E-01						
670	3.78E-01	1040	6.82E-01						
680	3.89E-01	1050	6.90E-01						
690	4.01E-01	1060	6.98E-01						
700	4.11E-01	1070	7.06E-01						
710	4.22E-01	1080	7.14E-01						

Form 1005-01 Rev E Page 2 of 2

We have added the related description and calculated equation in Page 19 Method “The IPCE measuring system (Beijing Zolix, Solar Cell Scan 100) consists of Xenon lamp and monochromator light. Before measurement, the monochromatic light intensity of the system is corrected with a standard silicon solar cell provided with the IPEC system. The IPCE efficiency calculation formula is shown in the formula, where λ represents the incident light wavelength, I_{light} represents the photocurrent density under illumination, I_{dark} represents the current density under dark field conditions, and P represents the light intensity under the corresponding wavelength.”

$$IPCE = \frac{\left[\frac{1240}{\lambda} \times (I_{light} - I_{dark}) \right]}{P} \times 100\%$$

We have revised related description in Page 9 Paragraph 3 “The incident photon-to-current efficiency (IPCE) of our device was measured at a potential of 1.5 V vs. RHE. By integrating the measured IPCE over the standard AM 1.5G spectrum (ASTM G173-03), a photocurrent density of 32 mA/cm² could be calculated (Supplementary Fig. 9d), which is very close to that of 34 mA/cm² at 1.5 V vs. RHE from the J-V test (Fig. 3a) under AM 1.5G simulator. The consistency between the two sets of photocurrent density demonstrates the accuracy of our AM 1.5G simulator in simulating sunlight, ensuring the reliability and accuracy of J-V and IPCE measurements.”

And we also added reference: [48] Boettcher SW, Spurgeon JM, Putnam MC, et al. Energy-conversion properties of vapor-liquid-solid-grown silicon wire-array photocathodes. *Science*, **327**: 185-187 (2010).

[49] Zhou X, Liu R, Lewis NS, et al. 570 mV photovoltage, stabilized n-Si/CoOx heterojunction photoanodes fabricated using atomic layer deposition. *Energy & Environmental Science*, **9**, 892-897 (2016).

Specific Comment R2-2: *Figure 9(d) in the supplementary information appears to be sourced from the dataset provided by the manufacturer of their solar simulator, without any accompanying reference to verify its origin.*

Response: We thank the reviewer for the precious comment and the suggestion. According to the suggestion, we have replaced the illumination data of the solar simulator to a new Supplementary Fig. 9 (d) for the IPCE with the integrated photocurrent density, where the integration was conducted for IPCE at each wavelength against standard AM 1.5G spectrum (ASTM G173-03, <https://www.nrel.gov/grid/solar-resource/spectra-am1.5.html>). Upon integration, a photocurrent density of 32 mA/cm² could be calculated, which is very close to that of 34 mA/cm² at 1.5 V vs. RHE from the J-V test (Fig. 3a) under our AM 1.5G simulator. The agreement in photocurrent density from both sets of measurements demonstrates the reliability of our AM 1.5G simulator in replicating sunlight.

The calculation method (*Energy Environ. Sci.* 2016, **9**, 892-897) is shown as follows:

$$I_{\text{Integrated}} = \int_{\lambda=300}^{\lambda} \frac{q}{hc} \text{IPCE}(\lambda) \cdot E(\lambda) \cdot \lambda \cdot d\lambda$$

Where E is the irradiance in W·m⁻²·nm⁻¹, h is the Planck's constant, c is the light speed.

Supplementary Fig. 9 (a) The digital photograph and (b) the schematic diagram of the PEC cell. (c) The specific position of three-electrode on the lid. (d) The IPCE measured at 1.5 V vs. RHE with integrated current of photoanode against standard AM 1.5G spectrum (ASTM G173-03).

We have added the related description and calculated equation in Page 19 Method “The ingetration of photocurrent based on the IPCE over standard AM 1.5G spectrum (ASTM G173-03) is shown as follows⁴⁹:

$$I_{\text{Integrated}} = \int_{\lambda=300}^{\lambda} \frac{q}{hc} \text{IPCE}(\lambda) \cdot E(\lambda) \cdot \lambda \cdot d\lambda$$

Where E is the irradiance in $\text{W} \cdot \text{m}^{-2} \cdot \text{nm}^{-1}$, h is the Planck’s constant, c is the light speed.”

Specific Comment R2-3: *In the experimental section, it is advisable to include information about the dopants used and their respective doping levels.*

Response: We acknowledge the helpful suggestion of the reviewer. We supply more information about dopants preparation on Method.

As suggested by the reviewer, we revised the related description in the Method “p-type a-Si was deposited using a gas mixture containing 4% silane diluted in hydrogen and BF_3 at 0.9 Torr and 40 W RF power for 200 s. n-type a-Si layer was deposited at 1.0 Torr, and 40 W RF power for 200 s, using a gas mixture containing 6.7% silane diluted

in hydrogen with the $\text{PH}_3:\text{SiH}_4$ doping gas.”

And we added related reference “[47]Shen L, Meng F, Liu Z. Roles of the Fermi level of doped a-Si: H and band offsets at a-Si: H/c-Si interfaces in n-type HIT solar cells. *Solar Energy*, 2013, 97: 168-175.”

Reviewer #3

General Comment R3: *The authors have addressed all the comments and the manuscript can be accepted now.*

Response: We thank the reviewer for the valuable comments.

REVIEWERS' COMMENTS

Reviewer #2 (Remarks to the Author):

The authors have appropriately addressed the feedback and comments given by the reviewer.